# Renal staffs' understanding of patients' experiences of transition from peritoneal dialysis to in-centre haemodialysis and their views on service improvement: A multi-site qualitative study in England and Australia

Janet E. Jones[ID][1]*, Sarah L. Damery[ID][1], Kerry Allen[2], David W. Johnson[ID][3¤a¤b], Mark Lambie[ID][4], Els Holvoet[5], Simon J. Davies[4]

1 Institute of Applied Health Research, University of Birmingham, Edgbaston, Birmingham, United Kingdom, 2 Health Services Management Centre, University of Birmingham, Edgbaston, Birmingham, United Kingdom, 3 Department of Nephrology, Princess Alexandra Hospital, Brisbane, Australia, 4 Faculty of Medicine and Health Sciences, Keele University, Staffordshire, United Kingdom, 5 Renal Division, Ghent University Hospital, Ghent, Belgium

¤a Current address: Australasian Kidney Trials Network, University of Queensland, Brisbane, Australia
¤b Current address: Translational Research Institute, Brisbane, Australia
* j.e.jones@bham.ac.uk

**Data Availability Statement:** Even if de-identified, the study data cannot be publicly shared due to

## Abstract

### Introduction

Many studies have explored patients' experiences of dialysis and other treatments for kidney failure. This is the first qualitative multi-site international study of how staff perceive the process of a patient's transition from peritoneal dialysis to in-centre haemodialysis. Current literature suggests that transitions are poorly coordinated and may result in increased patient morbidity and mortality. This study aimed to understand staff perspectives of transition and to identify areas where clinical practice could be improved.

### Methods

Sixty-one participants (24 UK and 37 Australia), representing a cross-section of kidney care staff, took part in seven focus groups and sixteen interviews. Data were analysed inductively and findings were synthesised across the two countries.

### Results

For staff, good clinical practice included: effective communication with patients, well planned care pathways and continuity of care. However, staff felt that how they communicated with patients about the treatment journey could be improved. Staff worried they inadvertently made patients fear haemodialysis when trying to explain to them why going onto peritoneal dialysis first is a good option. Despite staff efforts to make transitions smooth, good continuity of care between modalities was only reported in some of the Australian hospitals where, unlike the UK, patients kept the same consultant. Timely access to an appropriate service,

concerns over participant confidentiality, privacy, and the terms of participant consent, as noted by the REC that approved the study (London Bridge Research Ethics Committee (Ref: 18/LO/0974). Excerpts of interview transcripts relevant to the study are available on request from the research governance office of the University of Birmingham (researchgovernance@contacts.bham.ac.uk). Data have not been uploaded to a secure repository because of the approval conditions set out by the ethics committee.

**Funding:** This work was undertaken as part of the INTEGRATED consortium, funded by the Baxter Global Grant Program awarded to DWJ, ML and SJD (no associated grant number). SLD and JEJ were supported by the National Institute for Health Research Collaboration for Leadership in Applied Health Research and Care West Midlands (NIHR CLAHRC WM), now recommissioned as NIHR Applied Research Collaboration West Midlands (NIHR ARC WM) (https://nihr.ac.uk/). The views expressed are those of the authors and not necessarily those of the NIHR or the Department of Health and Social Care. DWJ is supported by an Australian National Health and Medical Research Council Leadership Investigator Grant (https://www.nhmrc.gov.au/funding/find-funding/investigator-grants). The study was sponsored by University of Birmingham. The funders and sponsors had no role in study design, data collection and analysis, decision to publish, or preparation of the manuscript.

**Competing interests:** The authors have declared that no competing interests exist.

such as a psychologist or social worker, was not always available when staff felt it would be beneficial for the patient. Staff were aware of a disparity in access to kidney care and other healthcare professional services between some patient groups, especially those living in remote areas. This was often put down to the lack of funding and capacity within each hospital.

## Conclusions

This research found that continuity of care between modalities was valued by staff but did not always happen. It also highlighted a number of areas for consideration when developing ways to improve care and provide appropriate support to patients as they transition from peritoneal dialysis to in-centre haemodialysis.

## Introduction

Treatments for people with kidney failure include peritoneal dialysis (PD), in-centre haemo-dialysis (HD), home haemodialysis (HHD), transplantation or conservative care. The majority of people with kidney failure will, at different stages of their disease, transition between several different treatment modalities [1,2]. PD first has been shown to have advantages for patient survival outcomes [3,4] and cost benefits for the healthcare provider [5]. Approximately a third of people on PD will transition to a different modality, most commonly to HD [6], and usually within three years [7]. This is often due to peritoneal access problems or infection [8]. Changing treatment modality, especially when unplanned, can be physically and psychologi-cally demanding, cause upheaval in patients' personal lives and lead to some experiencing con-siderable distress [9–11]. Whilst there is a growing evidence base about patient experiences of transitioning between treatment modalities [12,13], there is a paucity of evidence on healthcare professionals' perspectives on the transition process. Nevertheless, the current literature implies that transitions between modalities are poorly coordinated resulting in increased mor-bidity and mortality [14,15]. Understanding healthcare professionals' views of the barriers and facilitators of successful transitions from PD to HD, and their views on how clinical practice can be improved is essential to ensuring optimal service provision and patient care [7,16,17].

As part of the INTEGRATED consortium this international multi-site qualitative study aimed to develop in-depth understandings of staff perceptions of how patients experience transitions from PD to HD, barriers and facilitators to a successful transition, and staff views on how clinical practice might be improved [15]. The COREQ checklist (S1 File COREQ checklist) guided the reporting of this research [18].

## Methods

The previously published study protocol outlines the methods used in the INTEGRATED study [15]. Briefly, semi-structured interviews and focus groups were undertaken at three study sites in the West Midlands, United Kingdom (UK) and three sites in Queensland, Aus-tralia with the aim of identifying similarities and differences between participating sites and countries. Hospital sites in the UK and Australia were chosen because the models of care are generally different with regards to the practitioners involved in patient care before and after renal replacement therapy and the degree to which patients experience care continuity when moving between modalities. The sites in the UK and Australia were chosen to capture

variation in terms of the size of their PD programmes, the characteristics of their PD patients (age/ethnicity) and their catchment areas (rural/urban and socioeconomic characteristics of patients). All three UK hospitals were from different healthcare trusts and two of the Australian hospitals belong to the same healthcare trust. Ethical approval was obtained from the Health Research Authority (Ref: 237901) and London Bridge Research Ethics Committee (Ref: 18/LO/0974) in May 2018. Research Governance approval was obtained from each participating site in the UK and Australia.

## Participants and recruitment

Kidney care staff and allied health professionals were eligible to participate if they worked directly with patients on PD or HD in specialist teams; on acute hospital wards, or in haemodialysis units. All participants must have worked for their employee for a minimum of three months at the time of recruitment. The authors approached the clinical lead at each study site to ask if they were willing to act as a gatekeeper for the study. Gatekeepers identified and gauged interest in study participation of as many eligible staff as possible, aiming to achieve a sample of all relevant disciplines (e.g. nurses, junior doctors, consultants, healthcare assistants, dietitians, psychologists or any other appropriate staff members) in order to capture varying views and perspectives. Each interested member of staff gave permission for their contact details to be shared with the research team. Recruitment packs containing an invitation letter, participant information sheet and consent form were sent out via post or e-mail to all interested staff members, and all staff members who wished to participate were contacted to arrange an interview or focus group. Sampling took place prior to data collection, however capacity was built into the study design to allow further sampling of participants if required. All interviews and focus groups took place face-to-face at the hospital sites. The researchers were not acquainted with the participants prior to the interviews or focus groups.

## Data collection

All interviews and focus groups were conducted by the same two experienced qualitative researchers (one female and one male) in both the UK and Australia (KA and KS), who were qualified to PhD and Master's level, respectively, and were employed by the University of Birmingham, UK. Both researchers had experience of conducting qualitative research with patients with kidney disease and staff. No one else was present at the interviews and focus groups other than the researchers and the participants. Focus groups were our preferred method of data collection because of the discussions between stakeholders that focus groups can generate [19]. However, in a few cases, staff capacity meant we interviewed some members of staff individually. At the start of each interview or focus group, the researchers introduced themselves and explained the purpose of the research study. Both researchers took brief field notes after each interview and focus group. The topic guides sought to understand staff perspectives on patients' experiences of transition from PD to HD; how they feel patients cope; how staff and kidney care units can support patients and their families, and staff views about priorities for improving kidney care services related to treatment transitions (see S2 File interview topic guide and S3 File focus group topic guide). Repeat interviews or focus groups were not carried out and participants did not have the opportunity to comment on the transcript of their interview or focus group or to provide feedback on the findings. All participants provided written informed consent. All interviews were audio recorded and independently transcribed verbatim. To ensure accuracy, transcripts were proof-read against the recordings.

## Data analysis

Data saturation was reached after the six focus groups and three interviews in Australia and the one focus group and sixteen interviews in the UK after which any further sampling or data collection were not required. Data analysis commenced after all interviews and focus groups were completed. Transcripts were uploaded to NVivo 12 Plus [20] where data were analysed thematically and themes generated inductively without *a priori* expectations of findings [21]. An initial framework was developed using 10% of all the transcripts [22], this framework was then used to code the remaining transcripts. Data that did not fit this framework were discussed within the team and amendments or new codes generated accordingly until all data were analysed. Ten percent of transcripts were independently coded by two researchers (JJ and SLD). Disagreements were discussed by the coders and where agreement could not be reached the coding was discussed with the wider team. The relevant sections of the transcripts were reviewed and discussed until there was a unanimous decision about the most appropriate codes to use. Findings across the UK and Australia were synthesised to identify themes that transcended the differences between the two countries and those that were country specific.

## Results

Interviews lasted between 30–60 minutes and focus groups between 40–60 minutes. In total, sixty-one members of staff were recruited across the six sites (24 in the UK, 37 in Australia). Forty-two took part in focus groups (1 in the UK and 6 in Australia) and nineteen took part in interviews (16 in the UK and 3 in Australia). The number participants in a focus group ranged from 3 to 11 and each group had representation from a variety of job roles. The job roles of the participants are provided in Table 1.

Five themes were interpreted from the data: 1) communication between staff and patients, 2) care pathways, 3) access to kidney care services and other healthcare professionals, 4) staff skills and responsibilities and 5) service capacity. The majority of staff views were universal across all sites; where there were differences, these are described in the appropriate section. In the following quotes sites 1–3 refer to UK hospitals and 4–6 Australian hospitals.

### Communication between patients and staff

A good relationship between patients and staff was seen as essential for a smooth transition between treatment modalities. Nursing staff frequently explained how they like working in kidney medicine because it gives them the chance to build up a relationship with patients and their carers/family members. Staff felt that this relationship makes it easier for them to

**Table 1. Job roles of participants in interviews or focus groups.**

| Job role | Total number |
|---|---|
| Senior nursing role (incl. Advance nurse practitioner, consultant nurse, unit manager, sister) | 8 |
| Nursing staff (incl. healthcare assistant) | 30 |
| Consultant/Nephrologist | 13 |
| Psychologist | 4 |
| Dietitian | 2 |
| Social worker | 1 |
| Vascular access coordinator | 1 |
| Kidney care coordinator | 1 |
| Director of kidney care unit | 1 |
| **TOTALS** | **61** |

communicate information about renal treatments, especially about transition, through general conversations. Staff also explained how their relationship with the patients helped them to identify when patients were concerned about something:

> *"You start off with a patient and you think yeah it's fine, nothing to deal with there but by the time you've had a chat 'well actually'...."* (Site 3, Nurse).

Some members of staff explained how they had guidance on which topics to discuss with patients but no guidance on how best to communicate it. The importance of giving patients time to digest any information they are given and the opportunity to ask questions was highlighted:

> *"Have they actually understood? Have they taken it on aboard? How we are educating, is it effective, you know?......Can we be ensuring that we're getting their feedback? 'So tell me what you understand about what I have just told you?'.... and giving them permission to say 'I don't understand'".* (Site 6, Nurse).

Staff members (particularly in Australia) talked about how they thought the fear of transitioning to HD was greater now than years ago and described some patients as having an "anti-HD" mind set. Some members of staff in Australia thought that this may arise from the language used by health care professionals when talking about kidney failure and its treatments. Australia has a PD first policy and when initially discussing the benefits of PD with patients approaching kidney failure, staff felt that they may unintentionally imply that HD is the "last chance saloon." This may make patients fear HD when in reality, most patients who do not receive a kidney transplant relatively soon after commencing PD eventually need to transition to HD at some point during their treatment regimen:

> *"Even the terms we use, we say 'PD failure' which is a bit pejorative you know. I try wherever possible to use the term 'completion of PD' or something like that."* (Site 5, Director of kidney care unit).

## Care pathways

Staff talked about the wide range of emotions they thought patients experienced when advised they should consider transitioning to another treatment modality or conservative care. These include: anger at no longer being able to cope with PD, fear of the change and denial. Staff felt that having a planned transition provides patients with time to adjust to what will be their new way of life and in an ideal situation, they can start to prepare patients for transition as early as possible before the transition takes place. The gradual decline of health in those with renal disease means that often patients do not realise how ill they have become, so transition comes as a shock, is unsettling and patients are often unprepared which may result in patients disengaging with their treatment:

> *"The silent nature of kidney disease is probably the biggest issue because people don't realise that it's a problem until they start becoming symptomatic and once that happens, all of a sudden, it's a bit of a shock, but I think that's what contributes to a lot of the non-compliance, because they don't feel like they're sick until they're really, really sick."* (Site 6, Consultant nurse).

Doctors in Australia described how they typically looked after the same patients through their whole kidney journey regardless of treatment modality, although nursing staff are often

siloed within their dialysis specialty. However, the close proximity of the PD and HD units in some Australian hospitals meant that nursing staff would often check up on patients that have moved units offering recently transitioned patients reassurance and a friendly face. Staff in one Australian hospital talked about providing patients with holistic care in order to facilitate smoother more coordinated transitions. Although not all sites in Australia thought that this had been achieved:

> *"The way the service is run is just very medical, full stop, there isn't that integration of a different, not only holistic service but holistic person, patient."* (Site 6, Nurse).

In contrast, staff in the UK are, in general, compartmentalised into one kidney therapy modality making it difficult to provide continuity of care. However, this may be due to the renal practice model adopted by these specific UK hospitals and not representative of the model of care used in other UK hospitals. Recognising that improvement to patient care was required, some participants in the UK, when discussing how patient care could be improved, suggested that allocating a case manager to each patient may help to facilitate the continuity of care between dialysis modalities:

> *"The ideal would be that you have somebody, like a case manager who can go through the process."* (Site 1, Nephrologist).

There were conflicting views about how effective a case manager could be and not everyone was convinced this would be the best solution for the patients:

> *"But I think—as paradoxical as it would appear sometimes a fresh opinion changes things. . . . . .someone else walks in and says something else, it's like 'why didn't you say that?'". (Site 3, consultant).*

Finding that the process of transition was disjointed and that patients who needed to transition quickly were not accommodated for in the current system, staff at one site in the UK developed a transition service aimed at addressing the problem:

> *"I think you can plan for a proportion of patients. . .but the only way to manage most of these circumstances is to have some sort of transitional service".* (Site 3, consultant).

## Access to renal services and other healthcare professionals

Staff explained how understanding the practicalities associated with transitioning is important for patients and staff. The majority of practical issues are typically the same for patients in the UK and Australia, although attending a dialysis unit several times per week for treatment can pose greater challenges for Australian patients due to the distances that may be involved. For example, patients in the UK who are unable to travel to in-centre (hospital) dialysis units may be able to attend a nearby satellite unit. In contrast, in Australia, where patients may live hundreds of miles from a hospital or a satellite unit, PD undertaken at home is many patients' preferred treatment option due to its convenience. Patients living in remote rural locations may also be less affluent than those living in urban areas and the financial cost of moving nearer to the hospital is so great that patients are often unwilling to leave their communities when a transition to HD is medically advised. Patients may choose to have no further treatment rather than endure the expense and upheaval of moving:

*"It's diabolically difficult for them to move to what they regard as 'the big smoke' . . ..the poorest people would be lucky to buy a garage. . ..it's a terrible social problem."* (Site 4, Nephrologist).

All participants talked about the importance of providing patients with access to other healthcare professionals, such as psychologists, dieticians and social workers. There was a disparity across the participating hospitals in the availability of access to these services. For example, a member of staff at one UK hospital said:

*"We have very little access to Social Services, we have no access to Psychology."* (Site 2, Nephrologist).

A social worker in one Australian hospital explained how, although they do not see all transitioning patients, they often see those finding the transition to HD difficult because of their personal circumstances, demonstrating why the ability to refer to other appropriate healthcare professionals is so important:

*"If they're going to change from PD to haemo, often we only get involved if there are social issues that are very evident that may be the reason why they're failing in the sense that socially the support and everything isn't there to enable them to continue what they're doing."* (Site 4, Social worker).

Some patients are reluctant to transition and the reasons for this include feeling comfortable with PD; needle phobia; not wanting to come to the hospital regularly; family commitments, or they find HD to be unappealing. The following quote provides an example where access to either a social worker or psychologist or both may have provided a solution to the patient's needs:

*"She's a carer at home for her child's Downs Syndrome and I think really, really struggled with being you know, in hospital three times a week. . . ...just flat out, just could not, couldn't mentally accept it."* (Site 6, Nurse).

It is important to note that not all patients who are offered support from other healthcare professionals want it and this is particularly true of psychology. Needing psychological help can be seen as taboo or a stigma particularly in certain sections of society. Overall, staff believed that by having access to other healthcare professionals as a structured part of the kidney care pathway could help them to be proactive in addressing patient concerns about transitioning and treating any subsequent related social or mental health issues:

*"It's like when they've finally broken down in front of us that we see that, whereas if we did have it standardised, then you may mitigate some of that and be proactive."* (Site 5, Psychologist).

## Staff skills and responsibilities

Some nursing members of staff expressed an interest in learning about the kidney treatment modalities they do not work in and explained that they would find this helpful to provide a holistic view of potential treatment journeys to their patients:

*"I haven't got a clue what they do there [PD] at all, and I'd love to work in all areas because I think it's beneficial to be aware of exactly what's going on."* (Site 6, Nurse).

Although overall staff skills and wanting to learn about other modalities were not raised by the UK participants one described the introduction of a programme designed to give new staff members insight into all treatment modalities:

*"So the new recruitment process is, I think they're going to be based on the ward but they're going to spend time in CKD, time in PD, time in Haemo".* (Site 3, Nurse).

In Australia, staff often described confusion over who should be deciding which patients need to be transitioned from PD to HD when slots become available. Clinicians have sometimes delegated the decision to nurses on the basis that the nursing staff on the PD unit know the patients more intimately and will know who is or will be deteriorating rapidly. However, nurses think this should be a decision made by the clinicians. This difference in opinion has, at times, resulted in patients missing out on valuable HD slots:

*"So then the doctors won't make the decision and so the PD girls are like 'no, we're not making the decision', so the spot goes unfilled."* (Site 6, Nurse).

## Service capacity

Not all patients transitioning to HD want or are able to attend the in-centre unit at their nearest hospital. Satellite units can be more convenient for these patients in terms of location and/or treatment time slots, but satellite units have limited resources, are often full or may not have suitable slots available. Participants thought the provision of more regional dialysis units in Australia would overcome the need for patients to either move closer or travel hundreds of miles to an in-centre HD unit when they transition. Although tight budgets meant that need had to be weighed against cost-effectiveness:

*"They reckoned it wasn't cost-effective for a satellite unit for less than twelve. Warwick and Stanthorpe you could get the numbers, easily because of the big Indigenous population, but that's the dilemma. . ...of course it all costs money."* (Site 1, consultant).

Despite staff wishing for more joined up working between modalities, they acknowledged that available resources and funding will influence the services offered and how they operate. In some cases, they thought that this could be detrimental to a patient's health:

*"Because of the tightness of HD (available spaces) at the moment I feel like some of the long term patients probably would transition to HD but we're holding onto them a bit longer. So they are usually more acute."* (Site 5, Nurse).

## Discussion

To our knowledge this is the first multi-site international study researching the views of kidney care staff on patients' experiences of transition from PD to HD and how they would like to see clinical practice improved. Integrated care pathways and continuity of care for patients were important to staff and our research has highlighted differences in kidney care services between participating hospitals. Generally, the different kidney treatment modalities appeared to be separate from each other in the participating UK hospitals compared to a more integrated service model in some of the Australian hospitals. Participants identified a number of issues that can influence a patient's experience of transition but thought that patients having a planned

transition were less likely to experience upheaval compared to those who have an unplanned transition. This echoes the findings made elsewhere in this project in relation to patient experiences [12].

In the UK there was discussion about the introduction of a case manager role to help provide continuity of care for patients. This concurs with the Department of Health in England and Wales who recommend that each renal patient should be given a named contact who can coordinate care throughout their treatment journey [23]. However, it is currently difficult to recommend any particular model of care for renal patients due to a lack of evidence of their effectiveness [24].

The need for psychosocial support to be integrated into care pathways was discussed, particularly at the sites where, due to funding and capacity issues, they did not have regular and consistent access to psychological or social work support. The National Institute for Health and Care Excellence (NICE) guidance in the UK states that psychological evaluation, preparations and support should be provided prior to transition [25]. Similar to patient discussions reported in the paper by Allen et al staff explained how patients were resistant to transitioning for reasons such as family commitments, needle phobia and guilt [12]. The staff talked about how timely psychosocial support would be welcome for patients such as these because, as Seekles et al report, where psychosocial support is low, the needs of patients with kidney disease will often go unmet [26].

The provision of psychosocial support was viewed by staff to be one component of an effective integrated care pathway. Another aspect is ensuring that patients understand the treatment pathway and the reasons behind it. Education about renal disease and an understanding of treatment pathways has been reported as important to patients, and it is suggested that regular discussions between healthcare professionals, patients and their carers may help healthcare professionals to understand the impact that transitioning can have on the daily lives of patients and carers [12]. Recent studies have found that a lack of knowledge can lead to patient dissatisfaction and bad experiences with their kidney care [27,28] which may in turn lead to patients refusing or disengaging with their treatment [29]. Participants worried about the language they used, thinking it may make patients fear HD. To address this issue, the development of culturally appropriate educational materials, the use of consistent and easily understood terminology and practising shared understanding is recommended.

Access to the right services, in the right place and at the right time was raised as an issue by the majority of participants in both countries. The location and the accessibility of HD units is key to patients' acceptance of transition. In accordance with other research, we found that patients with kidney disease living in remote rural areas in Australia needed to either travel long distances or move house to be able to access the treatment they needed [27]. In their research, Thompson et al found that patients in the USA living more than 31 miles from a dialysis centre had increased mortality compared to those living closer to a centre [30]. In line with this, the Australian participants talked about how patients were more likely to comply with their treatment if there were more accessible satellite units in remote areas.

Although there can be different models of care in both the UK and Australia, all participants thought that patients valued continuity of care [31], although this was not specifically identified in patient interviews [12]. In the participating Australian hospitals it was more likely that patients would keep the same consultant when they transitioned from PD to in-centre HD whereas patients in the participating UK hospitals were more likely to change consultant. Overall, when different models of care were discussed the majority of participants, in both countries, thought that a treatment pathway where patients keep the same consultant was more beneficial for patients. Nursing staff expressed a desire to be trained in all treatment modalities so they would feel confident to discuss all possible treatments with their patients.

Upskilling nursing staff in this way will perhaps help to breakdown the perceived divisions between modalities.

## Strengths and limitations

The INTEGRATE consortium has also conducted and published research that sought patients' views of their transition from PD to in-centre HD [12]. Together, these two studies provide an overarching view of transition from PD to in-centre HD from both the staff and the patient perspectives. The number of staff participating in this research was high and represented a cross-section of job roles within kidney care units. However, we were reliant on gatekeepers to approach and screen suitable staff to take part in an interview or focus group. As part of the ethics approval the research team were only provided details of those staff members who were interested in taking part, therefore we are unable to provide numbers of staff approached and numbers of staff who declined to take part. The gatekeepers were unfortunately unable to secure interviews with UK allied health professionals (such as, dietitians and social workers), therefore the findings relating to individuals working in these roles only reflect the views from Australia. Whilst we aimed to recruit a mix of healthcare professionals we were unable to stratify by age, sex or ethnicity. Talking to staff within a short timescale only provides a snapshot of their views at that particular point in time, and does not account for any changes in clinical practice that may have happened since. The findings do not claim to be generalisable to all kidney care units as each will have its own practices. However, the individual issues and potential solutions discussed may be helpful either as a whole or in part to kidney care services in general.

## Conclusions

This research has highlighted a number of areas to be addressed when developing new care pathways for patients transitioning between PD and HD. These include: timely access to psychological and social work services, improved information sharing with patients about their potential treatment journey, continuity of care, and greater availability of kidney care services in remote rural areas. Although the participants agree that these service improvements will be beneficial for patients with kidney failure, they are conscious that extra funding and staff will be required and for resource poor health services this may be difficult to achieve. Therefore, clearer messages from policy makers about which aspects of kidney care services should be invested in to improve practice and patients' experiences will be welcomed.

## Supporting information

**S1 File. COREQ checklist.**
(PDF)

**S2 File. Interview topic guide.**
(DOCX)

**S3 File. Focus group topic guide.**
(DOCX)

## Acknowledgments

This study is part of the international INTEGRATED consortium with (in alphabetical order): C Chan, G Combes, S Davies, F Finkelstein, C Firanek, R Gomez, K Jager, V George, D Johnson, M Lambie, M Madero, I Masakane, D McDonald, M Misra, S Mitra, T Moraes, A

Nadeau-Fredette, P Mukhopadhyay, J Perl, R Pisoni, B Robinson, D Ryu, R Saran, J Sloand, N Sukul, A Tong, C Szeto, W van Biesen.

We would also like to acknowledge those who facilitated data collection in our study sites, particularly Sridevi Govindarajulu and Kenn-Soon Tan in Queensland, Australia and Kim Sein in the UK and Australia.

## Author Contributions

**Conceptualization:** Sarah L. Damery, Kerry Allen, David W. Johnson, Mark Lambie, Els Holvoet, Simon J. Davies.

**Data curation:** Kerry Allen.

**Formal analysis:** Janet E. Jones.

**Methodology:** Sarah L. Damery, Kerry Allen, David W. Johnson, Mark Lambie, Els Holvoet, Simon J. Davies.

**Writing – original draft:** Janet E. Jones.

**Writing – review & editing:** Sarah L. Damery, Kerry Allen, David W. Johnson, Mark Lambie, Els Holvoet, Simon J. Davies.

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
