## [Decision Letter · Decision Letter 0]

22 Apr 2021

PONE-D-21-06924

Renal staff understanding of patients’ experiences of transition from peritoneal dialysis to in-centre haemodialysis and their views on service improvement: A multi-site qualitative study in England and Australia

PLOS ONE

Dear Dr. Jones,

Thank you for submitting your manuscript to PLOS ONE. After careful consideration, we feel that it has merit but does not fully meet PLOS ONE’s publication criteria as it currently stands. Therefore, we invite you to submit a revised version of the manuscript that addresses the points raised during the review process.

We look forward to receiving your revised manuscript.

Kind regards,

Lucy E. Selman, BA, MPhil, PhD

Academic Editor

PLOS ONE

Journal Requirements:

2. Please include a copy of the interview guide used in the study, in both the original language and English, as Supporting Information, or include a citation if it has been published previously.

Reviewers' comments:

Reviewer's Responses to Questions

**Comments to the Author**

1. Is the manuscript technically sound, and do the data support the conclusions?

Reviewer #1: Yes

Reviewer #2: Yes

Reviewer #3: Yes

2. Has the statistical analysis been performed appropriately and rigorously? 

Reviewer #1: N/A

Reviewer #2: N/A

Reviewer #3: N/A

3. Have the authors made all data underlying the findings in their manuscript fully available?

Reviewer #1: No

Reviewer #2: No

Reviewer #3: No

4. Is the manuscript presented in an intelligible fashion and written in standard English?

Reviewer #1: Yes

Reviewer #2: Yes

Reviewer #3: Yes

5. Review Comments to the Author

Reviewer #1: I enjoyed reading this well written and interesting paper, presenting a well conducted study addressing a gap in understanding. I think the paper would benefit from a discussion comparing the findings from this study with those of patients’ perspectives, and I suggest the authors add this to the Discussion section. My detailed comments, questions and suggestions follow.

Abstract page 2 line 38 – The word ‘haemodialysis’ is used but later in Conclusions reference is made to ‘in-centre haemodialysis’. If the focus is in-centre HD then please specify this at the outset as transition to HD with a view to doing home HD may be different to transitioning to in-centre HD only. The rest of the paper reads as though in-centre HD is the focus.

The term patients is used throughout when ‘people’ might be suitable. ‘People with kidney failure’ may be a preferrable term.

Introduction page 4 lines 68/69: ‘Placing patients needing dialysis onto PD first’ – this sentence sounds quite paternalistic. ‘Placing’ suggests the clinical team chooses the treatment modality when this should be at least a shared decision. The phrase ‘needing dialysis’ is also problematic in that ‘dialysis need’ is not something about which there is consensus – with some people with kidney failure choosing conservative care with comparable life-expectancy and better quality of life. Suggest rephrasing this sentence.

Introduction page 4 lines 69/79: ‘and cost benefits’ – Please specify whether the cost benefits are for the healthcare provider and / or the patient.

Methods page 5 lines 92-94: ‘Hospital sites in the UK and Australia were chosen because of the different models of kidney care adopted in each of these countries’ – Please summarise the key differences in models of care. Please can you explain how the sites within the UK and within Australia were selected.

Methods page 5 line 105: You make reference to trying to achieve ‘a representative sample’ but this is a quantitative term, rarely appropriate for use in qualitative research. Qualitative research sampling is not usually intended to achieve ‘a representative sample’ but to achieve a diverse sample capturing all possible different perspectives/views/experiences. In addition there is only one characteristic mentioned with respect to sampling (job/position). Was any attempt made to achieve diversity with respect to e.g. gender, age etc. How was the sample size determined? You mention achieving data saturation but was that determined during data collection and resulted in sampling stopping? Overall further details are required regarding the sampling strategy.

Methods page 6 lines 117/118 – Should this sentence ‘were qualified to PhD and Master’s level’ read ‘WHO were qualified to PhD and Master’s level’ ?

Methods: Could you expand on how the sampling, data collection and analysis were related? Were they contemporaneous? Iterative? For example was analysis undertaken of a small number of interviews/focus groups, and subsequent sampling determined by early analysis? Or was sampling undertaken, interviews/focus groups completed, and then analysis?

Methods page 6 line 143: Were there any differences in coding/themes identified? How were any disagreements resolved?

Methods page 7 Data analysis: This section requires clarification. Were all interviews/focus groups inductively coded initially or was a subset of interviews/focus groups inductively coded to generate a framework, which was then used to code the remaining interviews/focus groups? Lines 140-141 please be specific about the % of transcripts used to generate the coding framework – was this 10%? Or 15%?

Methods: You’ve included the COREQ checklist but not written in the methods that this report was written with reference to this checklist. Please add this.

Methods: There is no discussion of non-participants. What proportion of invited individuals agreed to participation? Were any group of professionals more or less likely to participate compared to others?

Results page 8 Table 1: Is there a risk of identification of those individuals who were the only ones recruited from their professional group? e.g. Social worker, Vascular access coordinator, Kidney care coordinator, and Director of kidney care unit.

Results page 8 line 156: Remove capital letter from ‘Access’ to ensure consistency with how other themes are presented.

Results page 9: Table 2 doesn’t add anything extra to what’s already written in the text. Would it be possible to generate a thematic diagram that illustrates how the themes interact and maybe illustrates the differences between the UK and Australia samples?

Results page 9 line 172: Replace ‘its’ with ‘it’s’

Results page 10 lines 198-203: I’m not sure how this text and quote really fits under the theme.

Results page 10 line 206: ‘patients’ experienced’ should be ‘patients experienced’.

Results page 10 lines 206/207: I find the phrasing ‘when told they need to transition to HD’ problematic. It sounds like patients are instructed to transition to HD, when in reality patients have the choice to not transition, to opt for conservative care, or to try to bring forward a possible living donor transplant.

Results page 11 line 220: Please clarify what is meant by the term ‘Consultants’. In this sentence it sounds like you are describing doctors only. If so please specify, as the above quote is by a Consultant Nurse.

Results page 11 line 232: ‘In contrast, staff in the UK are, in general, compartmentalised into one kidney therapy modality’ – this sentence isn’t true for all hospitals in the UK. Practice patterns vary significantly. In the centre I work in for example, people with kidney disease stay under a single consultant’s care and are not treated by different consultants depending on their dialysis modality. I therefore think this is not really a difference between the UK and Australia, but rather the centres that you sampled in the UK and Australia. I think that therefore you should rewrite this to say that this difference is between the different models of practice, which may be in place in the UK and in Australia.

Results page 15 line 316: Replace ‘needs’ with ‘need’

Discussion: How do the staff perspectives compare to patients perspectives? Are the issues staff have identified as important/challenging etc the same as those identified by patients themselves?

Discussion page 17 line 363: Replace ‘psycho social’ with ‘psychosocial’

Discussion page 18 from line 387: As indicated above it is not correct to state that practice in the UK is to change consultant when changing treatment modality. This is not the case at all UK units. This section requires rewriting.

Reviewer #2: This is a very clearly written and easy to follow manuscript. Well done and thanks. The only thing I would value would be a little more information relating to the methodological orientation that underpins the study and its relevance to the generation of themes and comparisons drawn between the two settings. All other limitations of the study and manuscript are addressed in the appropriate sections by the authors.

Reviewer #3: This is a valuable analysis of renal staffs’ perceptions of patient experiences of transitioning between two renal treatment modalities. This is a well-written manuscript, focusing on the context in England and Australia.

My main area of concern/query relates to the method and analysis of data. I didn’t get a strong sense of what the underpinning methodological orientation was: this may be mentioned in the referenced protocol but would benefit from mention here (I had a brief read through ref 15 and couldn’t see a mention of thematic analysis there either?). It would be helpful to mention what type of thematic analysis was done, and how this was done in a stepwise fashion e.g. if a coding framework was developed from reading 10-15% of all transcripts or just a selection of particularly rich transcripts, and how codes were used to build the themes. Were there subthemes? Fleshing this out would be helpful to understand exactly how the analysis was completed – at the moment this feels a little vague.

It would be helpful to add information about why/how the specific sites were chosen, and providing some information about the 3 sites in each country. E.g. are these the same hospital or trusts, or different?

Table 1: does nursing staff include roles like healthcare assistants?

It might be beneficial to include who was in each focus group by demographic and the size of focus groups.

Line 103: Perhaps mention that the clinical lead at each site acted as a gatekeeper (this is described but not explicit, but later reflected on in the limitations), and how they were first involved/contacted.

Page 15: Staff skills and responsibilities: Table 1 mentions differences between country contexts, but I think this theme presentation only discusses Australian aspects: would be helpful to outline both to be able to see what the differences may be (as in the pathways and access themes).

For the quotes used to illustrate themes, I wasn’t sure which no. sites referred to which country (e.g. if 1-3 are Australia, 4-6 England) – apologies if I have missed this somewhere, but this feels like it would be helpful info to have to tease out differences.

Line 355: “This echoes the findings made elsewhere in this project in relation to patient experiences.” – can you add a reference for this?

I think this manuscript would benefit from more contextual overview of how services generally operate in both countries, and how typical these sites may be (to add to the background/methods), this would help reflect on some of the findings. For example, line 276: in the UK sites where there is no access to services such as psychology is this because the service is overcapacity or non-existent? Likewise, if possible, I would be really interested to know if the idea of case managers specifically came from the Australian model from discussions during data collection, or if this is something independent that came up in focus groups/interviews (this may be hinted at from line 391, but would like to see this explicitly). Furthermore, is it possible to draw on any examples in the literature of English services that have used this model (I believe there are some services that offer hybrid models or where patients stay with their usual consultant)?

Typos/grammar:

Title: renal staff understanding… I can see this both ways grammatically, but wonder if renal staffs’ understanding is clearer.

Line 156: capital on Access – is this needed?

Line 276: space needed between site2

Throughout something seems awry with the referencing: citations appear to be added after false stops, at the beginning of the next sentence, and often freestanding without a sentence e.g. line 365, 78, 81 etc. Would be beneficial to check placement of refs throughout (unless this is a ref style I am not familiar with).

Otherwise a good overview of the perceptions of renal staff!

6. PLOS authors have the option to publish the peer review history of their article (what does this mean?). If published, this will include your full peer review and any attached files.

Reviewer #1: No

Reviewer #2: **Yes: **Dr Barnaby Hole

Reviewer #3: No

---

## [Author Response · Author response to Decision Letter 0]

3 Jun 2021

Replies to editorial comments

Manuscript has been updated to meet PLOS ONE style requirements. 

Please include a copy of the interview guide used in the study, in both the original language and English, as Supporting Information, or include a citation if it has been published previously

Copies of both the interview and focus group topic guides are included. The topic guides are only available in English.

We note that the grant information you provided in the ‘Funding Information’ and ‘Financial Disclosure’ sections do not match. When you resubmit, please ensure that you provide the correct grant numbers for the awards you received for your study in the ‘Funding Information’ section

Thank you for providing us with the chance to update our funding information and financial disclosure. It should read: This work was undertaken as part of the INTEGRATED consortium, funded by the Baxter Global Grant Program awarded to DWJ, ML and SJD (no associated grant number). SLD and JEJ were supported by the National Institute for Health Research Collaboration for Leadership in Applied Health Research and Care West Midlands (NIHR CLAHRC WM), now recommissioned as NIHR Applied Research Collaboration West Midlands (NIHR ARC WM) (https://nihr.ac.uk/). The views expressed are those of the authors and not necessarily those of the NIHR or the Department of Health and Social Care. DWJ is supported by an Australian National Health and Medical Research Council Leadership Investigator Grant (https://www.nhmrc.gov.au/funding/find-funding/investigator-grants) . The study was sponsored by University of Birmingham. The funders and sponsors had no role in study design, data collection and analysis, decision to publish, or preparation of the manuscript.

If there are ethical or legal restrictions on sharing a de-identified data set, please explain them in detail (e.g., data contain potentially identifying or sensitive patient information) and who has imposed them (e.g., an ethics committee). Please also provide contact information for a data access committee, ethics committee, or other institutional body to which data requests may be sent.

We apologise if this was not made clear in the original submission. Even if de-identified, the study data cannot be publicly shared due to concerns over participant confidentiality, privacy, and the terms of participant consent, as noted by the REC that approved the study (London Bridge Research Ethics Committee (Ref: 18/LO/0974). Excerpts of interview transcripts relevant to the study are available on request from the research governance office of the University of Birmingham (researchgovernance@contacts.bham.ac.uk). Data have not been uploaded to a secure repository because of the approval conditions set out by the ethics committee.

Reviewer 1

I enjoyed reading this well written and interesting paper, presenting a well conducted study addressing a gap in understanding. I think the paper would benefit from a discussion comparing the findings from this study with those of patients’ perspectives, and I suggest the authors add this to the Discussion section. My detailed comments, questions and suggestions follow.

Thank you for reading our manuscript and taking the time to provide in-depth insightful comments. Similarities and differences between staff and patients are now highlighted in the discussion (page 19).

Abstract page 2 line 38 – The word ‘haemodialysis’ is used but later in Conclusions reference is made to ‘in-centre haemodialysis’. If the focus is in-centre HD then please specify this at the outset as transition to HD with a view to doing home HD may be different to transitioning to in-centre HD only. The rest of the paper reads as though in-centre HD is the focus.

We thank the reviewer for pointing this out. The focus of the study is to explore the transition from PD to in-centre dialysis. As suggested the abstract has been updated to reflect this. 

The term patients is used throughout when ‘people’ might be suitable. ‘People with kidney failure’ may be a preferrable term

The word patients was used throughout the manuscript as this reflects how the healthcare professionals referred to the people on kidney dialysis in the interviews and focus groups. However, we take on board the reviewer’s perspective and where appropriate we have changed patients to people.

Introduction page 4 lines 68/69: ‘Placing patients needing dialysis onto PD first’ – this sentence sounds quite paternalistic. ‘Placing’ suggests the clinical team chooses the treatment modality when this should be at least a shared decision. The phrase ‘needing dialysis’ is also problematic in that ‘dialysis need’ is not something about which there is consensus – with some people with kidney failure choosing conservative care with comparable life-expectancy and better quality of life. Suggest rephrasing this sentence.

Thank you for this insight. This sentence has now been revised (page 4). 

Introduction page 4 lines 69/79: ‘and cost benefits’ – Please specify whether the cost benefits are for the healthcare provider and / or the patient.

This refers to cost benefits to the healthcare provider and the manuscript now reflects this (page 4).

Methods page 5 lines 92-94: ‘Hospital sites in the UK and Australia were chosen because of the different models of kidney care adopted in each of these countries’ – Please summarise the key differences in models of care. Please can you explain how the sites within the UK and within Australia were selected.

Apologies for the lack of clarity about this. Hospitals were chosen to illustrate different models of care before and after renal replacement therapy and continuity of care for patients moving between modalities. UK and Australian hospitals varied in the size of their PD programmes, their catchment areas and the socioeconomic status of their patients. We have revised the methods section of the manuscript to describe these details (page 5). 

Methods page 5 line 105: You make reference to trying to achieve ‘a representative sample’ but this is a quantitative term, rarely appropriate for use in qualitative research. Qualitative research sampling is not usually intended to achieve ‘a representative sample’ but to achieve a diverse sample capturing all possible different perspectives/views/experiences. In addition there is only one characteristic mentioned with respect to sampling (job/position). Was any attempt made to achieve diversity with respect to e.g. gender, age etc. How was the sample size determined? You mention achieving data saturation but was that determined during data collection and resulted in sampling stopping? Overall further details are required regarding the sampling strategy.

Thank you for highlighting this. We agree this was an incorrect use of language. This section has been revised and now discusses how varying views and perspectives were sought by including a diverse sample of participants with differing relevant job roles (Page 6). We were unable to achieve diversity on age or gender because we were reliant on the participants identified by gatekeepers (See strengths and limitations section). Participants were sampled prior to data collection. Data saturation was achieved during data collection following which any additional sampling was not required. This information has been added to the data analysis section of the manuscript (page 7).

Methods page 6 lines 117/118 – Should this sentence ‘were qualified to PhD and Master’s level’ read ‘WHO were qualified to PhD and Master’s level’ ?

The reviewer is correct and we have amended this sentence in the manuscript (page 6). 

Methods: Could you expand on how the sampling, data collection and analysis were related? Were they contemporaneous? Iterative? For example was analysis undertaken of a small number of interviews/focus groups, and subsequent sampling determined by early analysis? Or was sampling undertaken, interviews/focus groups completed, and then analysis?

We agree this is not clear in the manuscript. Data analysis commenced after data collection was completed. See answer above (page 7).

Methods page 6 line 143: Were there any differences in coding/themes identified? How were any disagreements resolved?

We apologise for omitting this information in our manuscript. The coders discussed any disagreements in the coding between themselves. If an agreement could not be reached the coding was discussed with the wider team (page 8).

Methods page 7 Data analysis: This section requires clarification. Were all interviews/focus groups inductively coded initially or was a subset of interviews/focus groups inductively coded to generate a framework, which was then used to code the remaining interviews/focus groups? Lines 140-141 please be specific about the % of transcripts used to generate the coding framework – was this 10%? Or 15%?

We agree that this section is not clear. 10% of the interviews and focus groups were inductively coded and from these a framework was generated. This framework was then used to code the other transcripts (page 7).

Methods: You’ve included the COREQ checklist but not written in the methods that this report was written with reference to this checklist. Please add this.

We have now added a sentence to this effect (page 5).

Methods: There is no discussion of non-participants. What proportion of invited individuals agreed to participation? Were any group of professionals more or less likely to participate compared to others?

Thank you for raising this. Unfortunately the gatekeepers recruiting suitable participants were unable to provide us with this information due to ethical constraints on releasing any information (even de-identified details or numbers approached) to the research team without consent. In the UK gatekeepers were unable to recruit healthcare professionals such as dieticians and social workers. This information is provided in the strengths and limitations section (page 21).

Results page 8 Table 1: Is there a risk of identification of those individuals who were the only ones recruited from their professional group? e.g. Social worker, Vascular access coordinator, Kidney care coordinator, and Director of kidney care unit.

We agree with the reviewer that there may be a risk of identification. Therefore table 1 has been amended to show total number of participants per job role only rather than by country (page 8).

Results page 8 line 156: Remove capital letter from ‘Access’ to ensure consistency with how other themes are presented.

Capital A changed to an a.

Results page 9: Table 2 doesn’t add anything extra to what’s already written in the text. Would it be possible to generate a thematic diagram that illustrates how the themes interact and maybe illustrates the differences between the UK and Australia samples?

We agree that table 2 does not add anything to the narrative therefore we have decided to remove it from the manuscript. We have attempted to generate a thematic diagram, but this was similarly repetitive of what was already covered in the text, so we chose not to include a diagram.

Results page 9 line 172: Replace ‘its’ with ‘it’s’

Corrected (page 10).

Results page 10 lines 198-203: I’m not sure how this text and quote really fits under the theme.

Agreed. We have removed this quote (page 11).

Results page 10 line 206: ‘patients’ experienced’ should be ‘patients experienced’.

Corrected (page 11).

Results page 10 lines 206/207: I find the phrasing ‘when told they need to transition to HD’ problematic. It sounds like patients are instructed to transition to HD, when in reality patients have the choice to not transition, to opt for conservative care, or to try to bring forward a possible living donor transplant.

Thank you for this valuable insight. We have reworded this sentence to reflect that there is a choice of treatment modalities patients can transition to (page 11).

Results page 11 line 220: Please clarify what is meant by the term ‘Consultants’. In this sentence it sounds like you are describing doctors only. If so please specify, as the above quote is by a Consultant Nurse.

We apologise we are referring to a doctor here and this is now reflected in the manuscript (page 12).

Results page 11 line 232: ‘In contrast, staff in the UK are, in general, compartmentalised into one kidney therapy modality’ – this sentence isn’t true for all hospitals in the UK. Practice patterns vary significantly. In the centre I work in for example, people with kidney disease stay under a single consultant’s care and are not treated by different consultants depending on their dialysis modality. I therefore think this is not really a difference between the UK and Australia, but rather the centres that you sampled in the UK and Australia. I think that therefore you should rewrite this to say that this difference is between the different models of practice, which may be in place in the UK and in Australia.

This section is now revised to explain that in general staff in the UK are confined to one discipline but this is not always the case and other UK hospitals may adopt different practice models (page 13).

Results page 15 line 316: Replace ‘needs’ with ‘need’

Corrected (page 16). 

Discussion: How do the staff perspectives compare to patients perspectives? Are the issues staff have identified as important/challenging etc the same as those identified by patients themselves?

Staff and patient views are, where appropriate, now interwoven into the discussion.

Discussion page 17 line 363: Replace ‘psycho social’ with ‘psychosocial’

Corrected (page 19).

Discussion page 18 from line 387: As indicated above it is not correct to state that practice in the UK is to change consultant when changing treatment modality. This is not the case at all UK units. This section requires rewriting.

We have rewritten this section to reflect the fact that there can be different models of care in both the UK and Australia (page 20).

Reviewer 2

This is a very clearly written and easy to follow manuscript. Well done and thanks. The only thing I would value would be a little more information relating to the methodological orientation that underpins the study and its relevance to the generation of themes and comparisons drawn between the two settings. All other limitations of the study and manuscript are addressed in the appropriate sections by the authors.

Thank you for reading our manuscript. We hope that the issues around methodology have been addressed in our responses to comments from the other reviewers. Additional information has been added into the methods, participants and recruitment and data analysis sections. These include information on: the sampling of participants; sampling completed prior to data collection; data saturation achieved during data collection; data analysis took place after data collection was completed; how disagreements in coding were resolved; 10% of manuscripts were coded and a framework developed; the framework was then used to code the remaining transcripts. The discussion now includes comparisons with the patient interviews results paper.

Reviewer 3

My main area of concern/query relates to the method and analysis of data. I didn’t get a strong sense of what the underpinning methodological orientation was: this may be mentioned in the referenced protocol but would benefit from mention here (I had a brief read through ref 15 and couldn’t see a mention of thematic analysis there either?). It would be helpful to mention what type of thematic analysis was done, and how this was done in a stepwise fashion e.g. if a coding framework was developed from reading 10-15% of all transcripts or just a selection of particularly rich transcripts, and how codes were used to build the themes. Were there subthemes? Fleshing this out would be helpful to understand exactly how the analysis was completed – at the moment this feels a little vague.

Thank you for taking the time to read our manuscript and to provide insightful feedback. In response to reviewer 1 we have further described our methods and analysis in the manuscript. See the methods, participants and recruitment and data analysis sections for the revised text. These include information on: the sampling of participants; sampling completed prior to data collection; data saturation achieved during data collection; data analysis took place after data collection was completed; how disagreements in coding were resolved; 10% of manuscripts were coded and a framework developed; the framework was then used to code the remaining transcripts. Further limitations to the methods are outlined in the strengths and limitations section. There were initially sub-themes however following a wider team discussion it was decided that for clarity of message about experiences and service design these should be collapsed into the main themes. 

It would be helpful to add information about why/how the specific sites were chosen, and providing some information about the 3 sites in each country. E.g. are these the same hospital or trusts, or different? Table 1: does nursing staff include roles like healthcare assistants?

This information is now provided in the manuscript (page 5). Table 1: nursing staff does include healthcare assistants, the table now reflects this.

It might be beneficial to include who was in each focus group by demographic and the size of focus groups

We were unable to obtain demographic information on the participants as this may have increased the likelihood that participating individuals could be identified (see strengths and limitations). In the results a sentence about the general size of the focus groups and the participants in the focus groups is now included (page 8).

Line 103: Perhaps mention that the clinical lead at each site acted as a gatekeeper (this is described but not explicit, but later reflected on in the limitations), and how they were first involved/contacted.

The authors initially approached the clinical lead at each site to act as a gatekeeper for the study. This is now reflected in the manuscript (page 6).

Page 15: Staff skills and responsibilities: Table 1 mentions differences between country contexts, but I think this theme presentation only discusses Australian aspects: would be helpful to outline both to be able to see what the differences may be (as in the pathways and access themes).

Learning about other treatment modalities was not raised by UK staff. However there was discussion about new members of staff learning how the different modalities work and is now included in this section (page 16).

For the quotes used to illustrate themes, I wasn’t sure which no. sites referred to which country (e.g. if 1-3 are Australia, 4-6 England) – apologies if I have missed this somewhere, but this feels like it would be helpful info to have to tease out differences.

Please accept our apologies for this. A sentence in the results section now makes this clearer (page 9). 

Line 355: “This echoes the findings made elsewhere in this project in relation to patient experiences.” – can you add a reference for this?

Reference now added (page 18). 

I think this manuscript would benefit from more contextual overview of how services generally operate in both countries, and how typical these sites may be (to add to the background/methods), this would help reflect on some of the findings. For example, line 276: in the UK sites where there is no access to services such as psychology is this because the service is overcapacity or non-existent? Likewise, if possible, I would be really interested to know if the idea of case managers specifically came from the Australian model from discussions during data collection, or if this is something independent that came up in focus groups/interviews (this may be hinted at from line 391, but would like to see this explicitly). Furthermore, is it possible to draw on any examples in the literature of English services that have used this model (I believe there are some services that offer hybrid models or where patients stay with their usual consultant)?

Thank you for giving us the opportunity to expand on this. The methods section now explains that both UK and Australian hospitals can operate different models of care and the models of care generally (but not always) differ regarding the staff involved in patient care before and after renal replacement therapy and the continuity of patient care. The UK and Australian hospitals varied in the size of the PD patient community, patient socioeconomic characteristics and catchment areas (page 5). It was in the UK interviews and focus groups that case managers were mentioned, and were not related to the Australian model. We undertook a search of the literature but were unable to find any relevant papers on the different models of service provided in England. A discussion about this and how there is a lack of evidence on the effectiveness of different models of care is now included in the discussion (page 18).

Title: renal staff understanding… I can see this both ways grammatically, but wonder if renal staffs’ understanding is clearer.

We agree and the title now reflects this. 

Line 156: capital on Access – is this needed?

Capital A now changed to a.

Line 276: space needed between site2

Corrected.

Throughout something seems awry with the referencing: citations appear to be added after false stops, at the beginning of the next sentence, and often freestanding without a sentence e.g. line 365, 78, 81 etc. Would be beneficial to check placement of refs throughout (unless this is a ref style I am not familiar with).

Thank you for bringing this to our attention. They referencing is incorrect and has now been amended throughout the manuscript.

---

## [Editor Report · Decision Letter 1]

7 Jul 2021

Renal staffs' understanding of patients’ experiences of transition from peritoneal dialysis to in-centre haemodialysis and their views on service improvement: A multi-site qualitative study in England and Australia

PONE-D-21-06924R1

Dear Dr. Jones,

We’re pleased to inform you that your manuscript has been judged scientifically suitable for publication and will be formally accepted for publication once it meets all outstanding technical requirements.

Kind regards,

Lucy E. Selman, BA, MPhil, PhD

Academic Editor

PLOS ONE
---

## [Editor Report · Acceptance letter]

9 Jul 2021

PONE-D-21-06924R1 

Renal staffs’ understanding of patients’ experiences of transition from peritoneal dialysis to in-centre haemodialysis and their views on service improvement: A multi-site qualitative study in England and Australia. 

Dear Dr. Jones:

I'm pleased to inform you that your manuscript has been deemed suitable for publication in PLOS ONE. Congratulations! Your manuscript is now with our production department. 

Kind regards, 

on behalf of

Dr. Lucy E. Selman 

Academic Editor

PLOS ONE